# SOHET: Sequence Of Heterogeneous Events Transformer with Self-Supervised Pre-Training

Kees Jan de Vries * [1]   Mustafa Radha * [1]   Mathijs de Jong * [1]

## Abstract

Many machine learning applications rely on heterogeneous event streams to make predictions, either causally as events arrive or bidirectionally over complete sequences. We propose SOHET (Sequence Of Heterogeneous Events Transformer), a hierarchical architecture combining event-type-specific tabular encoders with temporal and type embeddings, processed by a causal or bidirectional transformer. We introduce three self-supervised pre-training objectives for the causal setting. On a proprietary large-scale real-world Booking.com fraud detection task with 17 event types, SOHET outperforms FlexTPP, NAPPT, and CIPPT by 5.8%. Pre-training yields an additional 2.6% gain and $2.4\times$ faster convergence. On the EBES benchmark, bidirectional SOHET matches or exceeds the published best on 6 out of 8 tasks.

## 1. Introduction

Across industries, many machine learning applications rely on heterogeneous event streams to make predictions over time for a given entity. In e-commerce, a user's browsing behaviour, purchases, account activity, etc. may be used for personalised recommendations, fraud detection, and other tasks (Chen et al., 2019; Moreira et al., 2021; Padhi et al., 2021). In healthcare, electronic health records contain diverse clinical events, such as laboratory measurements, medication administrations, and diagnoses, which are used to predict outcomes such as clinical deterioration, mortality, or readmission (Pang et al., 2021; McDermott et al., 2023). Similar settings arise in financial services, telecommunications, and cybersecurity. In this paper our primary focus is fraud detection at Booking.com, based on a proprietary

real-world dataset comprising 17 different event types.

Sequence modelling spans a wide variety of prediction settings. Besides continually updated predictions (requiring causal modelling), predictions can also be required once over a complete sequence (allowing bidirectional modelling). To address both settings for heterogeneous events, which subsumes the homogeneous case, we propose SOHET (Sequence Of Heterogeneous Events Transformer): a hierarchical transformer-based architecture combining temporal, type, and event-type-specific tabular encoding with causal or bidirectional sequence modelling.

While much pre-training work focuses on bidirectional modelling and/or homogeneous events, the causal heterogeneous case remains relatively underexplored. For our causal heterogeneous setting, we introduce next event time-difference and next event type prediction tasks, along with a novel contrastive-learning-based task on the encoding of tabular data.

Our contributions:

**(1) Strong supervised performance on causal heterogeneous events:** SOHET substantially outperforms open-source methods (FlexTPP (Draxler et al., 2025), NAPPT and CIPPT (McDermott et al., 2023)) on our proprietary large-scale real-world Booking.com fraud detection task with 17 event types.

**(2) Self-supervised pre-training for causal heterogeneous events:** SOHET further improves performance and convergence speed on the same Booking.com task through self-supervised pre-training, widening the gap over existing methods.

**(3) Strong supervised performance on sequence classification with homogeneous events:** SOHET achieves competitive performance on the EBES benchmark (Osin et al., 2025).

## 2. Related Work

Work on sequential modelling over event data can be organized along two axes central to the applications described above. First, whether predictions are updated continu-

---

*Equal contribution [1]Booking.com, Amsterdam, The Netherlands. Correspondence to: Kees Jan de Vries <kees.devries@booking.com>.

ally as events arrive or made once from a complete sequence, respectively making *causal* and *bidirectional* sequence modelling a natural fit. Second, whether the sequence contains a single event type (*homogeneous*) or multiple (*heterogeneous*) event types, each with its own set of attributes. Bidirectional homogeneous modelling includes TabBERT (Padhi et al., 2021) and FATA-Trans (Zhang et al., 2023). Bidirectional heterogeneous approaches include CEHR-BERT (Pang et al., 2021) and UniTTab (Luetto et al., 2025). Causal homogeneous modelling has been explored through TabGPT (Padhi et al., 2021), autoregressive transaction pre-training (Skalski et al., 2023), and transformer-based recommender systems (Chen et al., 2019; Moreira et al., 2021; Xia et al., 2023; Li et al., 2025). The causal heterogeneous setting, closest to ours, has been addressed through neural temporal point processes (Du et al., 2016; Zuo et al., 2020) and recent extensions: FlexTPP (Draxler et al., 2025) and EventStreamGPT (ESGPT) (McDermott et al., 2023), which includes the NAPPT and CIPPT model variants. SOHET combines temporal, type, and event-type-specific tabular encoding (Gorishniy et al., 2021; Borisov et al., 2024) in a single hierarchical architecture.

Our Fraud detection task requires a causal heterogeneous model. To the best of our knowledge, EventStreamGPT (McDermott et al., 2023), including its CIPPT and NAPPT variants, and FlexTPP (Draxler et al., 2025) are the only implementations of causal heterogenous sequence of events models. These works provide especially relevant baselines because their autoregressive forecasting objectives overlap directly with two of our three SOHET pre-training tasks: predicting the next event time and next event type. EventStreamGPT formulates this as generative modelling of complex events with event-internal structure: CIPPT assumes that intra-event covariates are conditionally independent given the history, whereas NAPPT relaxes this through a dependency-graph-based nested attention mechanism. FlexTPP similarly targets mixed-type event streams, using an autoregressive transformer with classification and regression heads. Where SOHET differs most clearly from these baselines is in how the next event contents are represented and learned during pre-training. We replace feature-level reconstruction with a contrastive objective on the latent representation of the next event.

## 3. SOHET Model Architecture

### 3.1. Problem Formulation

We consider sequences $\{e_1, e_2, \ldots, e_T\}$ of heterogeneous events $e_t = (\tau_t, s_t, \mathbf{x}_t)$, consisting of a timestamp $\tau_t$, an event type $s_t \in \mathcal{S}$ and tabular features $\mathbf{x}_t$. The feature schema depends on the event type: $\mathbf{x}_t \in \mathcal{X}_{s_t}$.

### 3.2. Event Encoding

Each event encoding is the sum of three components, as shown in Figure 1a: concatenated piecewise linear embeddings (Gorishniy et al., 2022) of the relative time difference $\Delta\tau_t^{\mathrm{r}} = \tau_t - \tau_{t-1}$ and cumulative time difference $\Delta\tau_t^{\mathrm{c}} = \tau_t - \tau_0$, producing $\mathbf{h}_t^{\mathrm{time}}$; a learned event-type embedding $\mathbf{h}_t^{\mathrm{type}}$; and a feature encoding $\mathbf{h}_t^{\mathrm{features}}$ obtained by passing event-type-specific tabular features through a per-type FT-Transformer (Gorishniy et al., 2021), whose [CLS] token maps all event types into a shared embedding space. The final encoding is $\mathbf{h}_t = \mathbf{h}_t^{\mathrm{time}} + \mathbf{h}_t^{\mathrm{type}} + \mathbf{h}_t^{\mathrm{features}}$.

### 3.3. Sequence Modelling

The sequence of event encodings $\{\mathbf{h}_1, \mathbf{h}_2, \ldots, \mathbf{h}_T\}$ is processed by a position-aware sequence model and outputs a sequence of event representations $\{\mathbf{h}_1', \mathbf{h}_2', \ldots, \mathbf{h}_T'\}$, as shown in Figure 1b. For tasks requiring a bidirectional sequence model, like per-sequence classification, we use a ModernBERT encoder (Warner et al., 2025). ModernBERT provides practical advantages over other BERT-like models, like unpadding and Flash Attention. Unpadding is particularly beneficial when sequences vary in length. For causal tasks, we use a ModernBERT decoder (Weller et al., 2026), which applies causal masking to ensure representations at position $t$ depend only on events $1, \ldots, t$.

### 3.4. Self-Supervised Pre-Training

We propose three complementary objectives that leverage the natural structure of event sequences, as shown in Figure 1c. These pre-training objectives require a causal sequence model, since they all involve predicting future information. The total pre-training loss is the sum of these three objectives.

**Next Event Type Prediction.** Given the representation $\mathbf{h}_t'$, we predict the type $s_{t+1}$ of the next event using a classification head over $|\mathcal{S}| + 1$ classes (event types plus an end-of-sequence indicator).

**Next Time Delta Prediction.** Given the representation $\mathbf{h}_t'$, we predict when the next event will occur by regressing both the relative and cumulative transformed time deltas: $\log_{10}\big((\Delta\tau_{t+1}^{\mathrm{r,c}} + 1\,\mathrm{second}) \,/\, 1\,\mathrm{day}\big)$.

**Contrastive Next-Row Embedding Learning.** Inspired by JEPA (Assran et al., 2023), we predict the encoding of upcoming tabular features in latent space rather than reconstructing raw features. Where JEPA avoids representation collapse through an asymmetric architecture, we instead use contrastive learning: a learned projection head maps the current event representation $\mathbf{h}_t'$ to a query vector whose cosine similarity with the feature encodings of the next $M$

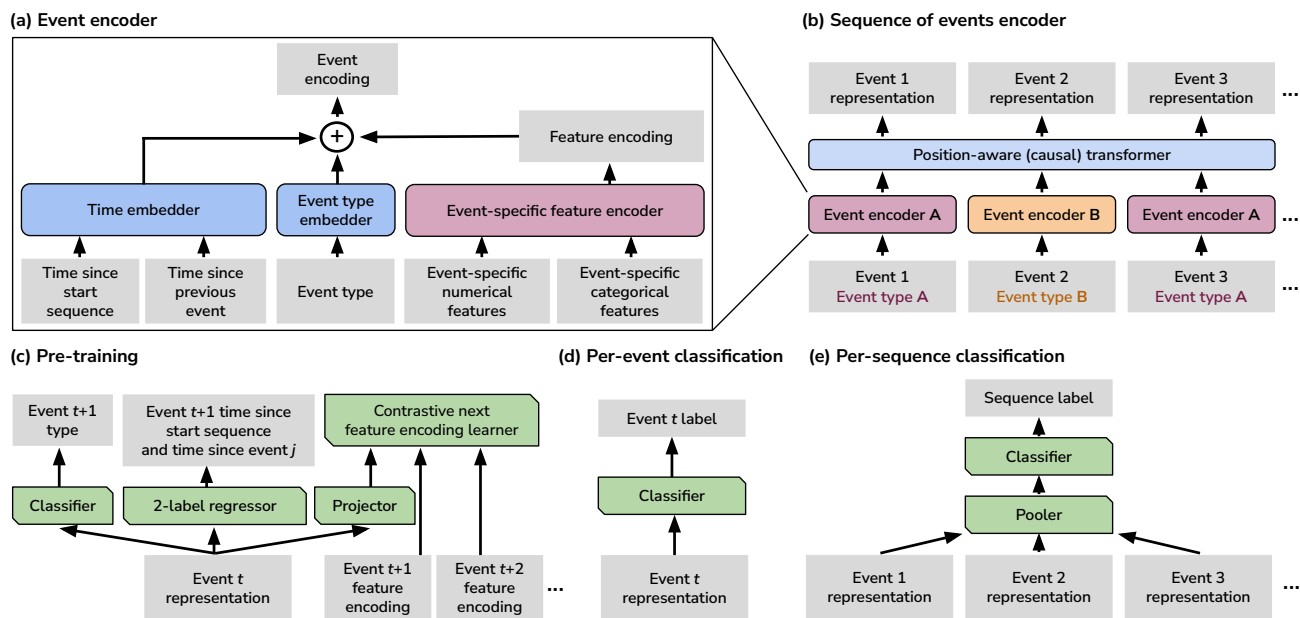

*Figure 1.* SOHET architecture. (a) Event-specific tabular encoders transform heterogeneous events into event encodings. (b) Event encodings are processed by a position-aware sequence encoder. The resulting event representations can be used for various tasks. (c) Our novel self-supervised objectives include next event type prediction, next time delta prediction, and contrastive next-row embedding learning. (d) Per-event classification and (e) per-sequence classification are typical downstream tasks relevant for real-life scenarios. See Appendix A for full details.

events is computed as:

$$\text{sim}_{t,k} = \frac{\text{Proj}(\mathbf{h}'_t)^T \mathbf{h}^{\text{features}}_{t+k}}{\|\text{Proj}(\mathbf{h}'_t)\| \|\mathbf{h}^{\text{features}}_{t+k}\|} \tag{1}$$

where $k \in \{1, \ldots, M\}$. Following CLIP (Radford et al., 2021), these similarities are scaled by a learned log-temperature $\tau$, normalised with softmax, and trained with cross-entropy loss targeting $k = 1$ (the immediate next event).

### 3.5. Downstream Task Adaptation

Our framework is highly flexible and supports diverse downstream tasks. For per-event tasks, we place a task-specific head on each event representation (Figure 1d). For per-sequence tasks, we pool event representations for a task-specific head (Figure 1e). We address two main settings.

**Per-event causal prediction** predicts an event-level target $y_t$ using only past context up to event $e_t$: $p(y_t \mid \{e_1, \ldots, e_t\})$ with $t <= T$. Our Fraud prediction task, which requires updated predictions at each new event arriving in a stream, is an example of this setting. The causal constraint during training guarantees that the model does not learn from future events and can therefore be deployed for real-time inference. The labels $y_t$ can be constant for a given sequence. The model can be trained supervised from scratch, or it can first be pre-trained using our self-supervised objectives followed by supervised fine-tuning.

**Per-sequence classification** predicts a single label $y$ for the entire sequence: $p(y \mid \{e_1, \ldots, e_T\})$. This applies when the full sequence is available at inference time, such as classifying completed user sessions. This task typically benefits from a bidirectional sequence model.

## 4. Experiments

**Heterogeneous Event Sequences.** We evaluate SOHET on the proprietary **Booking.com Fraud Detection Dataset**: a large-scale real-world dataset of heterogeneous accommodation-supplier timelines comprising 17 event types, including property onboarding, reservations, customer service interactions, financial transactions, content uploads, and reviews. The task is online fraud-risk prediction, requiring the model to update its estimate as new events arrive, i.e. causal sequence modelling. Sequences are constructed from 1 year of historical data with a time-based test split; the training set is further split 80/20 into train and validation subsets. For details, see Appendix B.1.

SOHET uses hidden dimension 320 with 4 FT-Transformer blocks and 4 ModernBERT layers (8 transformer blocks total). We compare against FlexTPP (Draxler et al., 2025), CIPPT, and NAPPT (McDermott et al., 2023), discussed in Section 2. All baselines were matched to SOHET's hidden dimension and transformer depth for fair comparison. As a result, SOHET has more trainable parameters, because the SOHET architecture includes event-specific encoders.

*Table 1.* Test average precision (AP) on the Booking.com Fraud Detection Dataset, reported as mean $\pm$ standard deviation over five seeds.

| Method | Supervised | SSL + FT |
|---|---|---|
| SOHET (ours) | **0.452 ± 0.005** | **0.464 ± 0.007** |
| FlexTPP | 0.414 ± 0.008 | 0.440 ± 0.007 |
| NAPPT | 0.427 ± 0.003 | 0.417 ± 0.007 |
| CIPPT | 0.406 ± 0.004 | 0.414 ± 0.005 |

For pre-training, the three self-supervised objectives are weighted equally. See Appendix B for full hyperparameter details.

**Homogeneous Event Sequences.** We also validate SOHET's sequence modelling capabilities on the EBES benchmark (Osin et al., 2025), which includes 10 datasets spanning diverse domains and provides strong baselines for comparison. We use 8 of the 10 datasets; MIMIC-III was excluded due to access restrictions and MBD due to computational constraints. Since these tasks involve sequence classification with full context, we use bidirectional attention. SOHET hyperparameters were tuned per dataset using Optuna's Tree-structured Parzen Estimator (TPE) sampler (Akiba et al., 2019; Bergstra et al., 2011) (50 trials) over hidden dimension, attention heads, and layer counts. Results from 9 baseline models were reused from the EBES benchmark (Osin et al., 2025). See Appendix C for dataset and baseline details.

## 5. Results

**Fraud Detection Results.** Table 1 presents results on the Booking.com Fraud Detection Dataset. In the supervised setting, SOHET achieves 0.452 AP, outperforming the strongest baseline NAPPT (0.427) by 5.8%. This demonstrates the value of hierarchical architecture for heterogeneous events even without pre-training; Appendix B discusses possible reasons.

With pre-training, SOHET achieves 0.464 AP, a 2.6% improvement over supervised-only SOHET and 5.4% over the best pre-trained baseline (FlexTPP at 0.440). Fine-tuning reaches peak validation AP about 2.4× faster than training from scratch. Pre-training degrades NAPPT and CIPPT: both models' generative losses exhibit numerical instability with gradient norms diverging during training. Further tuning may mitigate this.

**EBES Benchmark Results.** Table 2 shows performance across EBES. SOHET matches or exceeds the published best on 6 out of 8 tasks, with the clearest gain on Pendulum (+7.9%). It is competitive but not best on PhysioNet2012 and ArabicDigits. The strong Pendulum result is consistent with EBES time-importance analyses (see Appendix C).

*Table 2.* Performance on EBES benchmark datasets. Best baseline is the max across 9 models from Osin et al. (2025). SOHET results are mean $\pm$ standard deviation over five seeds and match or exceed the published best on 6 out of 8 datasets. See Appendix C for full comparison and more details.

| Dataset | Best baseline | SOHET (ours) |
|---|---|---|
| Taobao | 0.713 | **0.715 ± 0.001** |
| BPI17 | 0.754 | **0.759 ± 0.003** |
| PhysioNet2012 | **0.846** | 0.818 ± 0.008 |
| Retail | 0.553 | **0.555 ± 0.004** |
| Age | **0.636** | **0.636 ± 0.003** |
| ArabicDigits | **0.986** | 0.984 ± 0.003 |
| ElectricDevices | 0.741 | **0.744 ± 0.006** |
| Pendulum | 0.777 | **0.838 ± 0.017** |

## 6. Discussion and Limitations

Our results validate SOHET's three contributions. First, event-type-specific encoders prove effective on Booking.com fraud detection, outperforming competing causal heterogeneous event models (5.8% gain). The modular architecture scales to rich event schemas. Second, self-supervised pre-training improves AP and reaches peak validation performance about 2.4× faster, evidence that pre-training produces useful representations. Third, competitive EBES performance demonstrates that hierarchical design combining tabular and sequential modelling works on established benchmarks, matching or exceeding the published best on 6 out of 8 datasets.

These results suggest SOHET is a useful backbone for structured event models; further validation on public heterogeneous event datasets would strengthen this claim. Ablation studies could clarify the importance of architectural components. Future encoding improvements include parameter sharing across event types with overlapping schemas and multi-hash embeddings for rare categories such as identifiers. For security and fraud applications, pre-training could be strengthened with out-of-sequence detection for causal models, and with closest-sequence classification and masked-event objectives in a bidirectional setting. Finally, the framework can be extended to multimodal inputs combining tabular with unstructured data types (e.g. vision, language).

## 7. Conclusion

We introduced SOHET, a hierarchical architecture combining event-type-specific tabular encoders with sequence-level temporal modelling that achieves strong performance on a proprietary large-scale fraud detection task and competitive performance on established benchmarks. The modular design makes SOHET a flexible backbone for modelling structured event sequences.

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

# A. Architecture Details

### A.1. Event Encoding Details

**Time Delta Embedder.** The relative time difference $\Delta\tau_t^r$ and cumulative time difference $\Delta\tau_t^c$ are each encoded using piecewise linear embeddings (Gorishniy et al., 2022) with 14 bins at the following boundaries: 1 second, 10 seconds, 1 minute, 30 minutes, 1 hour, 3 hours, 12 hours, 1 day, 1 week, 30 days, 90 days, 180 days, 52 weeks, and 520 weeks. The two resulting embeddings are concatenated to form $\mathbf{h}_t^{\text{time}}$.

**Event-Type-Specific Encoders.** Each event type has its own FT-Transformer encoder (Gorishniy et al., 2021):

- **Categorical features** are embedded using learned lookup tables. Vocabulary sizes are determined from training data; out-of-vocabulary values are mapped to a special UNK token.

- **Numerical features** are linearly projected with feature-specific weights and biases.

### A.2. Sequence Modelling Details

**Pooling.** For per-sequence classification tasks, we use masked mean pooling over event representations: the sequence representation is the mean of all non-padding event representations.

**Position-Aware Concatenation.** For batching variable-length sequences with heterogeneous events, we use position-aware concatenation. Each event in a sequence is assigned a global position index reflecting its temporal order. Events are first processed by their type-specific FT-Transformer encoders, then placed into a dense tensor using their global position indices. Padding positions use zero-valued embeddings, with a sequence mask indicating valid positions.

### A.3. Self-Supervised Pre-Training Details

**Contrastive Projection Head.** The projection head used in contrastive next-row embedding learning (Section 3.4) is a single-layer MLP: LayerNorm $\to$ ReLU $\to$ Linear, with both input and output dimensions equal to the hidden size.

**Contrastive Learning Configuration.** The contrastive objective considers $M = 10$ next rows. The temperature parameter is learned during training, initialized to 1.0. All three self-supervised losses are weighted equally.

### A.4. Preprocessing

Across all experiments, the same preprocessing method is used. Our preprocessing pipeline consists of a set of preprocessors for each event type. Each event preprocessor contains a numerical and categorical pipeline, which are fitted to that event type's respective features. Numerical pipelines perform simple constant-value imputation for missing value, followed up by a quantile transformer to normalize feature distributions to a standard distribution. The number of quantiles used for fitting is $min(min(|X|, 1e^6)/n_{bins}, max_{quantiles})$ with $n_{bins} = 30$, $max_{quantiles} = 1000$ and $|X|$ is the number of events available in the training data. The categorical pipelines are embedded, where all categories with less than 100 examples are reduced to a single category.

# B. Fraud Detection Experiment

### B.1. Dataset

We construct sequences from 1 year of historical data from an e-commerce platform. Each sequence represents the timeline of an accommodation supplier and comprises 17 event types spanning the supplier lifecycle, including property onboarding, reservations, customer service interactions, financial transactions, content uploads, and reviews. Each event type carries its own set of numerical and categorical features, ranging from a single categorical field (e.g. ticket type) to dozens of numerical and categorical attributes (e.g. onboarding risk signals).

We use a time-based test split; the training set is further split 80/20 into train and validation subsets. Timestamps are converted to seconds since sequence start. Pre-training uses all available sequences; supervised training uses sequences with labelled transactions.

## B.2. Evaluation Protocol

Fraud results are reported as mean and standard deviation over five random seeds. For SSL+FT runs, we pre-train once per method and repeat fine-tuning over the five seeds from the resulting checkpoint. Convergence speed is measured from the validation AP history.

## B.3. SOHET Configuration

**Architecture (fixed, not tuned):**

- FT-Transformer row encoder: hidden dimension 320, 16 attention heads, 4 blocks

- ModernBERT sequence encoder: hidden dimension 320, 16 attention heads, 4 layers

**Training:**

- Optimiser: AdamW with learning rate $1 \times 10^{-4}$ (constant schedule)

- Batch size: 128 sequences (both supervised and self-supervised)

- Maximum sequence length: 1024

- Early stopping patience: 20 epochs

## B.4. Baseline Configuration

All baseline models were matched to SOHET's hidden dimension of 320. To ensure a fair comparison, we matched transformer depth across models:

- **SOHET**: 4 FT-Transformer blocks + 4 ModernBERT layers = 8 transformer blocks, resulting in 79M trainable parameters

- **FlexTPP**: depth 8, resulting in 11M trainable parameters

- **NAPPT**: 4 StructuredBlocks $\times$ 2 TransformerBlocks each (sequence attention + dependency graph attention) = 8 transformer blocks, resulting in 10M trainable parameters

- **CIPPT**: 4 TransformerBlock layers, resulting in 5M trainable parameters

NAPPT is the most architecturally comparable baseline to SOHET, as both employ a hierarchical design with two attention stages. We matched NAPPT's capacity to SOHET's. CIPPT shares a framework with NAPPT (EventStreamGPT), so we kept its hyperparameters consistent with NAPPT. CIPPT's effective depth of 4 (versus 8 for the other models) is a known limitation; extending CIPPT to 8 layers is left for future work.

**FlexTPP-specific:** hidden dimension 320, $d_k = 20$, $d_{ff} = 1280$, dropout 0.0, learning rate $3 \times 10^{-5}$ (selected via grid search over $\{10^{-5}, 3 \times 10^{-5}, 10^{-4}, 3 \times 10^{-4}, 10^{-3}\}$).

**Shared training settings (FlexTPP, NAPPT, CIPPT):** batch size 32, maximum sequence length 1024, early stopping patience 20 epochs. For supervised training, NAPPT and CIPPT use learning rate $1 \times 10^{-4}$ (constant schedule); FlexTPP uses $3 \times 10^{-5}$ as noted above. For pre-training, NAPPT and CIPPT use learning rate $1 \times 10^{-5}$.

## B.5. Discussion

SOHET performs better than FlexTPP, NAPPT and CIPPT in our fraud detection experiment. We did not perform full ablation studies to understand which aspects of the SOHET architecture are mainly responsible for its high performance. Here we highlight some differences between SOHET and the baseline architectures that might be responsible for the performance difference.

- SOHET is a hierarchical model, with (1) event-type specific event encoders, and (2) a sequence encoder. The inclusion of the event-type specific encoders makes the model more expressive than the baseline models, which operate on flattened sequences.

- We use two ways to represent time ($\Delta\tau_t^r$ and $\Delta\tau_t^c$) instead of one. In addition, we apply piecewise linear embeddings to these representations; each of the 14 bins is assigned its own embedding, which makes the resulting time embedding very expressive.

- The categorical encoding of the baseline models is limited, because they are based on overlapping vocabularies over the different features.

## C. EBES Benchmark

### C.1. Datasets

The following datasets from the EBES benchmark (Osin et al., 2025) were used to produce the results for homogeneous sequences:

- Taobao (e-commerce clickstream)

- BPI17 (business process)

- PhysioNet2012 (healthcare time series)

- Retail (retail transactions)

- Age (age prediction from sensor data)

- ArabicDigits (temporal pattern recognition)

- ElectricDevices (device classification)

- Pendulum (physical system prediction)

The Multimodal Banking Dataset (MBD) and MIMIC-III dataset were not used. MIMIC-III required additional certification for access and the MBD dataset was too large to run the benchmark on.

### C.2. Baselines

We used the models included in the EBES benchmark (Osin et al., 2025) as strong baselines to compare our approach with. The benchmark includes the following models: CoLES (Babaev et al., 2022) (contrastive learning for event sequences), GRU (Cho et al., 2014) and MLEM (Moskvoretskii et al., 2024) (RNN-based sequence models), Transformer (Vaswani et al., 2017) (standard bidirectional), Mamba (Gu & Dao, 2024) (state-space model), ConvTran (Foumani et al., 2024) (convolutional-transformer hybrid), mTAND (Shukla & Marlin, 2021) and PrimeNet (Chowdhury et al., 2023) (models for irregularly-sampled time series).

### C.3. Experimental Setup

Hyperparameters were tuned per dataset using Optuna's TPE sampler (Akiba et al., 2019; Bergstra et al., 2011) (50 trials). The search space was:

- Hidden dimension: $\{64, 128, 192, 256, 384\}$

- FT-Transformer encoder attention heads: $\{4, 8, 16\}$

- ModernBERT attention heads: $\{4, 8, 16\}$

- FT-Transformer encoder layers: 1–4

*Table 3.* Performance comparison across EBES benchmark datasets (Osin et al., 2025). Metrics are ROC AUC for binary classification tasks (Taobao, BPI17, PhysioNet2012) and accuracy for multi-class classification tasks (Retail, Age, ArabicDigits, ElectricDevices, Pendulum). SOHET results are mean ± standard deviation over five seeds and match or exceed the published best on 6 out of 8 datasets.

| Method | Taobao | BPI17 | PhysioNet2012 | Retail | Age | ArabicDigits | ElectricDevices | Pendulum |
|---|---|---|---|---|---|---|---|---|
| CoLES | **0.713** | 0.742 | 0.840 | 0.553 | **0.634** | 0.983 | 0.729 | 0.740 |
| GRU | **0.713** | 0.754 | **0.846** | 0.543 | 0.626 | 0.975 | 0.741 | 0.683 |
| MLEM | **0.713** | 0.753 | **0.846** | 0.544 | 0.636 | 0.978 | 0.736 | 0.676 |
| Transformer | 0.692 | 0.549 | 0.838 | 0.536 | 0.621 | 0.986 | 0.710 | 0.658 |
| Mamba | 0.693 | 0.737 | 0.835 | 0.538 | 0.609 | 0.983 | 0.716 | 0.687 |
| ConvTran | 0.703 | 0.748 | 0.837 | 0.534 | 0.603 | 0.986 | 0.711 | 0.674 |
| mTAND | 0.672 | 0.738 | 0.841 | 0.519 | 0.582 | 0.951 | 0.631 | 0.777 |
| PrimeNet | 0.681 | 0.730 | 0.839 | 0.521 | 0.583 | 0.958 | 0.636 | 0.600 |
| MLP | 0.659 | 0.737 | 0.835 | 0.526 | 0.581 | 0.760 | 0.437 | 0.186 |
| **SOHET (ours)** | **0.715 ± 0.001** | **0.759 ± 0.003** | 0.818 ± 0.008 | **0.555 ± 0.004** | **0.636 ± 0.003** | 0.984 ± 0.003 | **0.744 ± 0.006** | **0.838 ± 0.017** |

- ModernBERT layers: 1–4

The hidden dimension was constrained to be divisible by the maximum of both attention head counts. Tuning was performed with batch size 64; final training used batch size 128 with learning rate $1 \times 10^{-4}$ (constant schedule), maximum sequence length 1024, and early stopping patience of 10 epochs. We randomly selected 15% of the training split as a validation dataset, used only for early stopping and hyperparameter tuning.

### C.4. Full Results

Table 3 shows the full results of the EBES benchmark. The results for the baseline models (all except SOHET) were taken directly from the EBES publication (Osin et al., 2025); baseline uncertainty estimates can be found in that paper. For the results on the SOHET model, we used the preprocessing scripts provided in the EBES github repository (https://github.com/On-Point-RND/EBES) to produce the sequences, targets and train/test splits. The test split was held out purely for determining the final performance numbers shown in Table 3. The SOHET results are reported as mean and standard deviation over five random seeds. The reported performance numbers are either ROC AUC for binary classification tasks (Taobao, BPI17 and PhysioNet2012) or accuracy for multi-class classification tasks (Retail, Age, ArabicDigits, ElectricDevices and Pendulum).

### C.5. Discussion

SOHET most strongly outperforms the baseline models on the Pendulum dataset, with smaller gains or ties on Taobao, BPI17, Retail, Age, and ElectricDevices, and no gains on PhysioNet2012 and ArabicDigits. The EBES authors investigated the importance of time in the EBES datasets and showed that time is most important in Pendulum, ArabicDigits and ElectricDevices and much less important in the other datasets. SOHET processes time in a more expressive way, using two methods to calculate time ($\Delta\tau_t^r$ and $\Delta\tau_t^c$) and applies piecewise linear embeddings on these time differences. This is consistent with the strong Pendulum result, although the mixed results on ArabicDigits and ElectricDevices indicate that time encoding alone does not explain performance across all EBES datasets.

