# OpenReview forum: "SOHET: Sequence Of Heterogeneous Events Transformer with Self-Supervised Pre-Training"
_ICML.cc/2026/Workshop/FMSD — FMSD @ ICML 2026 Poster_

### Official Review · Reviewer_gppp · 2026-05-20
**SOHET: Sequence Of Heterogeneous Events Transformer with Self-Supervised Pre-Training**

**Rating:** 9
**Confidence:** 4

**Review:**

The paper proposes a hierarchical architecture for making predictions based on heterogeneous event streams, called SOHET. Each event is converted to an embedding by a dedicated event encoder that combines a time embedder, an event type embedder, and an event-specific encoder. The resulting embeddings serve as input to a GPT-like transformer, which can be used for per-event or per-sequence classification or regression tasks. The embeddings are pretrained on tasks such as next event type prediction, next time delta prediction, and contrastive next-row embedding learning. SOHET is evaluated on a proprietary fraud detection dataset and the EBES benchmarks against existing methods such as FlexTPP, NAPPT, and CIPPT, demonstrating superior performance in many cases.

Given that heterogeneous event streams are widespread in many applications, SOHET addresses a highly relevant problem and shows modest improvements over existing solutions.

It would be valuable if the code for SOHET were made available to facilitate reproducibility and adoption.

---

### Official Review · Reviewer_S5sw · 2026-05-22
**Strong event-sequence architecture, but the evidence is hard to verify and not parameter-matched**

**Rating:** 6
**Confidence:** 3

**Review:**

### Summary

This paper proposes SOHET, a transformer model for heterogeneous event sequences where each event has a timestamp, an event type, and type-specific tabular features. The model uses event-type-specific FT-Transformer encoders, adds time and type embeddings, and then processes the event sequence with either a causal or bidirectional ModernBERT-style transformer. For the causal setting, the paper introduces three self-supervised pretraining tasks: next event type prediction, next time-delta prediction, and a contrastive objective over the next event's feature embedding. Experiments are run on a proprietary fraud detection dataset with 17 event types and on 8 EBES benchmark datasets.

### Strengths

- The problem setting is important and relevant to the workshop. Many real structured-data problems are not just static tables but sequences of heterogeneous events.
- The architecture is clean and reasonable. Using separate event-type encoders and then mapping all event types into a shared sequence space is a natural way to handle different schemas.
- The contrastive next-row objective is the most interesting part of the paper. Predicting the next event representation in latent space is a good alternative to reconstructing every raw feature.
- The reported fraud detection results are strong: SOHET improves over FlexTPP, NAPPT, and CIPPT, and pretraining further improves PR-AUC and convergence speed.
- The EBES results give some public benchmark support, with SOHET reaching the best result on 6 out of 8 datasets.

### Weaknesses

- The main result is on a proprietary dataset, so the strongest claim is difficult to verify independently. This is understandable for fraud detection, but it still limits how much weight can be put on the headline numbers.
- The comparison is not really parameter-matched. The appendix reports SOHET at 79M parameters, while FlexTPP, NAPPT, and CIPPT are much smaller. This makes it hard to separate architectural gains from capacity gains.
- The paper does not include ablations, which is a major issue here. The method combines event-specific encoders, expressive time embeddings, ModernBERT, and a new contrastive pretraining loss, but the reader cannot tell which part is actually driving the improvement.
- The pretraining claim is only convincingly tested on the proprietary task. On EBES, the paper mainly evaluates supervised bidirectional SOHET, so the public experiments do not really validate the causal pretraining contribution.
- The "foundation model backbone" framing feels somewhat overstated. The model is useful and flexible, but the paper does not show broad transfer, zero-shot behavior, or cross-domain pretraining in the usual foundation-model sense.

### Suggestions

The most important addition would be a smaller parameter-matched SOHET baseline, ideally around the same size as NAPPT/FlexTPP. I would also like to see at least one ablation separating the time encoding, event-type-specific encoders, and contrastive pretraining objective. A public heterogeneous event dataset with causal pretraining results would also make the main contribution much easier to judge.

### Justification of Score
I recommend marginal accept. The paper is relevant, practically motivated, and provides promising empirical results. However, the main evidence is proprietary, the comparison is not capacity-controlled, and the absence of ablations makes it difficult to attribute the gains. The work is likely to interest the FMSD audience, but it should be presented as a strong applied architecture rather than a fully validated foundation-model contribution.

---

### Official Review · Reviewer_Rcwx · 2026-05-22

**Rating:** 8
**Confidence:** 4

**Review:**

The paper introduces SOHET, a transformer-based architecture for heterogeneous event sequence modeling, targeting fraud detection across 17 event types. It uses type-specific encoders combined with temporal and type embeddings, and supports both causal and bidirectional modes. The authors also proposes three self-supervised pretraining objectives for the causal setting, improving performance and convergence speed.

Strengths:
- Strong empirical gains in fraud detection, outperforming prior approaches by a clear margin.
- Self-supervised pretraining significantly improves optimization speed and final performance.
- Competitive results extend beyond heterogeneous fraud data to homogeneous sequence classification tasks.

Weaknesses:
- Limited analysis of which components (type-specific encoders vs. pretraining objectives) contribute most to the gains.
- Weak discussion of generalization to other domains beyond fraud detection.
- Missing ablation or stress tests on robustness under distribution shift or rare event types.